# Integrated Climate Change Mitigation and Public Health Protection Strategies: The Case of the City of Bologna, Italy

**DOI:** 10.3390/ijerph21111457

**Published:** 2024-10-31

**Authors:** Isabella Nuvolari-Duodo, Michele Dolcini, Maddalena Buffoli, Andrea Rebecchi, Giuliano Dall’Ò, Carol Monticelli, Camilla Vertua, Andrea Brambilla, Stefano Capolongo

**Affiliations:** 1Design and Health Lab, Department of Architecture, Built Environment and Construction Engineering (DABC), Politecnico di Milano, 20133 Milan, Italy; isabella.nuvolariduodo@polimi.it (I.N.-D.); michele.dolcini@polimi.it (M.D.); maddalena.buffoli@polimi.it (M.B.); andrea.rebecchi@polimi.it (A.R.); stefano.capolongo@polimi.it (S.C.); 2Department of Architecture, Built Environment and Construction Engineering (DABC), Politecnico di Milano, 20133 Milan, Italy; giuliano.dallo@polimi.it; 3TextilesHub, Department of Architecture, Built Environment and Construction Engineering (DABC), Politecnico di Milano, 20133 Milan, Italy; carol.monticelli@polimi.it (C.M.); camilla.vertua@polimi.it (C.V.)

**Keywords:** health promotion, climate change resilience, wellbeing, public health, healthy cities

## Abstract

Introduction: The ongoing process of global warming, driven by the escalating concentration of greenhouse gases generated by human activities, especially in urban areas, significantly impacts public health. Local authorities play an important role in health promotion and disease prevention, and some aim to achieve net-zero greenhouse gas emissions. There is a consistent action underway to reach this goal, hence the need for mapping and implementing effective strategies and regulations. Materials and Methods: This study includes the analysis of policy guidelines adopted by the city of Bologna, consulted in March and April 2024. Bologna is one of the 100 cities committed to achieving climate neutrality by 2030, 20 years ahead of the EU target. To identify the strategies adopted to mitigate climate change, the following methodology was used: (i) the systematic mapping of sources and spatial planning documents; (ii) the extrapolation of goals, measures, and target indicators; and (iii) the development of an overall matrix. Results: The main findings of the study and their connection to public health pertain to the identification of key macro-areas contributing to the reduction of greenhouse gas emissions, while reducing the impact of climate change on health: (1) built environment and renewable energy sources, (2) transport and mobility, (3) energy, (4) green areas and land use, and (5) citizen support. Within these five macro-areas, 14 goals have been identified, to which a total of 36 measures correspond, and, finally, a target indicator is determined, mainly with respect to the reduction of tons of CO_2_ equivalent per year. Conclusions: In order to protect public health, it is evident that buildings and urban activities should not produce carbon emissions throughout their lifecycle. This paper presents a method to evaluate municipal policies regarding dual-impact solutions that address both environmental protection through sustainability strategies and public health, in compliance with the Health in All Policies (HiAP) approach.

## 1. Introduction

### 1.1. Theoretical Background

The ongoing process of global warming, due to the increased concentration of greenhouse gases (GHGs) caused by human activity, has a strong impact on public health [1]. The effects of climate change are being felt today, and future projections represent an unacceptably high and potentially catastrophic risk to human health [2].

The direct impacts of climate change cause higher temperatures, more frequent floods, droughts, and an increase in the frequency of severe storms. Indirectly, it poses significant risks to public health by intensifying air pollution, facilitating the spread of disease vectors, leading to food insecurity and malnutrition, causing displacement, and contributing to mental health issues. Currently, it is evident that there is a need to address the development of more sustainable lifestyles to reduce carbon footprint in urban areas and climate change, as well as the spread of diseases [3]. It is scientifically proven that noncommunicable diseases (NCDs), like cardiorespiratory and skin diseases, cancer, allergopathies, obesity, diabetes, stress, anxiety, sleeping disorders, cognitive development impairment, and social exclusion, are mainly connected to the environmental risk factor [4]. To prevent potentially catastrophic climate change impacts, it is crucial to limit the global average temperature rise to below 2 °C, which necessitates keeping total human-caused carbon dioxide (CO_2_) emissions under 2900 billion tons (GtCO_2_) by the end of the century [2,4]. One possible method of gathering support for carbon emissions reduction is to emphasize the added co-benefit of reducing present-day, health-related impacts [5].

Local authorities, from this perspective, play a key role in health promotion, disease prevention, and spatial planning. At the urban scale, this is increasingly crucial for the identification of climate change mitigation and resilience initiatives [6]. Resilience has become an increasingly important management priority and planning goal for cities, especially for climate change adaptation [7].

Urban environments are extremely vulnerable to the public health effects of global warming, posing a significant threat to the long-term sustainability of urban settlements development [8].

Urban areas are responsible for over 70% of global greenhouse emissions, proving to be the most significant burden in the global effort towards decarbonization. Among 167 major cities at the global level, analyzed by a study conducted in 2021, less than 25% had committed to achieving carbon neutrality targets [9].

### 1.2. Net-Zero City Mission in EU Context

The transformation of European cities towards zero-emission models and climate change adaptation has taken shape through numerous initiatives under the European Union’s Green Deal. Cities also represent a crucial cornerstone in climate change mitigation strategies at the EU level. With regard to the European Union context, most cities have defined an emission target in 2050 to align with the EU’s broader climate objectives.

Although urban areas make up only 4% of the total land, they are home to 75% of its citizens [10]. To accelerate efforts in urban contexts against climate change, the European Commission has launched this initiative aimed at supporting a selected list of 100 cities across all 27 EU member states and 12 cities from Horizon Europe associate countries, through dedicated funding and governance tools. The specific objective of the European project is to develop a resource- and water-efficient, climate-resilient economy and society, ensuring the sustainable management of natural resources and ecosystems, and a balanced use of raw materials to meet the needs of a growing global population within the planet’s sustainable limits.

The initiative has two primary objectives:To deliver 100 climate-neutral and smart cities by 2030;To ensure that these cities serve as hubs for experimentation and innovation, enabling all European cities to achieve similar goals by 2050.

The cities involved have been requested to draft “Climate City Contracts”, which outline comprehensive plans with specific objectives and strategies to achieve climate neutrality across various sectors, including energy, buildings, waste management, and transport, along with associated investment strategies. The documents are developed collaboratively through a stakeholder participative approach that involves citizens and local organizations. The commitments articulated in the “Climate City Contracts” allow cities to engage effectively with EU, national, and regional authorities, as well as with the private sector and investors and the communities, to achieve their climate neutrality goals.

Under this policy program, in 2022, the Italian City of Bologna was selected by the European Commission to achieve net-zero emissions of greenhouse gases, in order to protect public health [11]. Bologna was, therefore, considered as an application case study as it anticipates the activities and obligations of the European Green Deal [12]. In February 2024, after a public consultation with a community involvement process, the City of Bologna published the “Climate City Contract” [13].

### 1.3. The City of Bologna

In 2015, the inventory of emissions was updated to assess the trends of the city and monitor the effectiveness of what was implemented, recording a reduction of almost 300,000 tons of CO_2_, or a drop of 12.4%. This is a very positive result that outlines a new trend that will probably allow us to achieve the −20% target for 2020. At the same time, Bologna was one of the first cities in Italy to adopt a local plan for adaptation to climate change [14]. This Plan is the result of the BLUE AP project (Bologna Local Urban Environment Adaptation Plan for a Resilient City), funded by the LIFE+ program (LIFE11 ENV/IT/119), which the Municipality of Bologna coordinated between 2012 and 2015 involving Kyoto Club, Ambiente Italia, ARPAE Emilia Romagna, and CMCC (Euro-Mediterranean Center for Climate Change) [15]. The climate scenario in Bologna is shown in Figure 1.

### 1.4. Aim of the Paper

This study explores the enforcement of local policies aimed at reducing climate impacts and protecting urban population health through strategies that encompass various domains and sectors. In this scenario, researchers and practitioners, both of technical and medical education, identified the need for interdisciplinary work [16]. These urban planning tools and documents, drafted in recent years, share the common goal of developing environmental sustainability strategies and reducing climate-altering emissions at the urban scale. They need to be framed in order to guide coherent city planning in the future. Specifically, the objective of this study is to assess local planning tools in compliance with the European Commission’s NetZeroCities strategies for combating climate change, with the aim of achieving climate neutrality by 2030. Thus, this study aims to pinpoint, within the planning documents in a specific case study of a medium-sized city in northern Italy, the identified climate change mitigation strategies with climate neutrality goals and with the Health in All Policies (HiAP) approach.

Bologna’s climate data, as many other European cities, underscore the need for climate adaptation measures. Bologna is experiencing improvements in air quality, partially due to stricter emissions standards and expansion of pedestrian zones. Interventions in affordable housing and public services are needed to ensure equitable access for all socioeconomic groups.

## 2. Materials and Methods

This study follows several steps, which illustrate the methodological process leading to the drafting of the evaluation matrix focusing on sustainability and climate change in the context of the Bologna 2030 Climate Mission [17]. The methodological steps are shown in Figure 2. The study includes the analysis of policy guidelines adopted by the city of Bologna, consulted in March and April 2024 from the site of the Municipality and of the Metropolitan City of Bologna. The analysis was carried out according to the following methodology, articulated in stages.

### 2.1. Systematic Mapping of Urban Planning Documents

This stage is of particular importance as it represents the scientific basis of the research, fundamental for the elaboration of the matrix. In this phase, a comprehensive collection of all relevant planning documents and local policies that address sustainability and climate change was assembled. This included a thorough review of urban planning frameworks, strategic climate action plans, and sustainability policies that have been developed to tackle environmental challenges. The aim was to gather a complete set of documents that reflects the current local strategies and initiatives related to climate action, ensuring a robust foundation for subsequent analysis and evaluation. This collection process is crucial for understanding the conformity of local policies with broader climate goals. The analysis of guidelines, urban planning, and policy documents of the City of Bologna was carried out, with particular reference to strategic climate and sustainability objectives. In defining goals and indicators, first, a general consultation of tools related to the environmental aspect of the Municipality of Bologna was carried out; successively, the relevant documents were gathered and categorized into two main groups: urban planning tools and ongoing projects. While the first ones include formal documents and master plans and transportation plans, the second ones are more specific projects, detailing current infrastructure development or social initiatives being undertaken by the Municipality of Bologna. Sources are grouped and classified in Table 1.

### 2.2. Extrapolation of Goals, Actions, and Target Indicators

This phase aims to extract and identify key intervention areas for Environmental Sustainability and Climate Change Mitigation, starting with the environmental objectives defined in the Paris Climate Agreement and in the European Grean Deal [29]. Based on the identified macro-areas, a list of goals for the current scenario is defined, along with a list of measurable targets to assess the effectiveness of the implemented policies and actions. Based on the mission objectives and the contents of the abovementioned documentation, strategic goals are identified and grouped by environmental issues. This is followed by the identification of measures linked to the respective goal, to which a target indicator is determined.

### 2.3. Development of a Climate Action Matrix

The phase following the extrapolation of the goals and the identification of the relevant measures concerns the drafting of a matrix, which collects and returns the information described above in order to have an overall view of the data. It includes four columns, respectively: goal, measure, target indicator, source.

## 3. Results

### 3.1. Sources Mapping

Among the territorial urban planning tools, the following documents were considered: the Action Plan for Sustainable Energy and Climate (APSEC) [18], approved by the Municipality of Bologna in April 2021; the General Urban Plan (GUP) [19], which came into effect in September 2021; the Urban Plan for Sustainable Mobility (UPSM) [20], adopted in November 2018; and the Action Plan for Sustainable Energy (APSE) [21], approved by the City Council in May 2012.

### 3.2. Macro-Areas and Goals

The strategic guidelines of the Climate City Contract [13] laid the foundation for defining the five macro-areas listed below. They represent the most important issues concerning risk mitigation and climate change adaptation measures.

The above-mentioned documents represent the outcome of a selection among plans and projects of the city of Bologna related to energy–environmental issues, collected from the sites of the municipality and metropolitan city of Bologna. The documents considered cover the period from 2018 to 2023, in addition to an urban forestation project dating back to 2008. Figure 3 illustrates the relation between the sources and the goals. Below are listed the 14 goals that emerged from the projects and urban planning tools, grouped by macro-area, listed with letters.

A.Construction (Built Environment and Renewable Energy Sources)Encourage the regeneration of manmade land and contrast land consumption.Energy saving and energy efficiency in residential buildings.Supporting energy transition and circular economy processes.B.Transport and Mobility4.Mobility planning focused on cycling and walking.5.Strengthening metropolitan public transport.6.Redistribution of motorized mobility networks.7.Development of innovative mobility.C.Energy:8.Efficiency enhancement of public lighting installations.9.Control and regulation of lighting installations.10.Reductions in energy consumption, especially among particularly significant users.D.AFOLU (Agriculture, Forestry and Other Land Use)11.Enhancement of green areas and urban eco-networks.12.Preventing and mitigating environmental risks.13.Developing urban blue infrastructure.E.Supporting citizenship14.Training, awareness, inclusion.

### 3.3. Identification of Goals, Actions, and Target Indicators

Encourage the regeneration of manmade land and contrast land consumptionPreserving soil quality, promoting soil health, and restoring areas that have been damaged or depleted by human activities, while also trying to limit the net loss of soil due to factors such as unplanned urbanization or intensive agriculture.
*Recovering and improving the efficiency of the existing building stock:*
The Plan envisages, on the one hand, working on disused or underused buildings in the urban area, and, on the other hand, renovating the built heritage that is inadequate with respect to energy saving and seismic safety issues.
*Completing the parts of the city where the transformation is not complete:*
The Plan focuses not only on the energy requalification of existing buildings, but also on the optimization of urban spaces through the completion of areas undergoing urban transformation.
*Encouraging the reuse and urban regeneration of manmade areas:*
The Plan aims to improve urban resilience by reducing the consumption of nonrenewable resources and reclaiming urban soils. It encourages the transformation of existing urban fabric through building and urban planning interventions.Energy saving and efficiency in residential buildingsThis objective indicates the importance of adopting measures and practices to reduce energy consumption in buildings used for residential purposes. This implies the implementation of technologies, systems, and policies that reduce the consumption of energy used for heating, cooling, lighting, and other purposes within the residential building.
*Consolidating already existing functions:*
In order to enhance environmental and urban quality, the Plan allows limited interventions outside the boundaries of the urbanized territory only when they are necessary to enhance the functions already present.
*Define energy-efficiency interventions for buildings:*
The construction industry represents one of the main targets of energy saving policies. After 2020, new buildings must be nearly zero-energy buildings and a large part of the remaining consumption must come from renewable sources. The interventions with the greatest impact are aimed at improving the envelope and act on windows and doors, external wall insulation, and roof insulation.
*Promoting and incentivizing different forms of energy efficiency:*
The city of Bologna has the goal of reducing gas emissions to at least 40 per cent of the emissions measured in 2005. This requires energy requalification of the existing building heritage, which includes energy-saving measures, energy efficiency, and the implementation of zero-emission renewable energy sources.
*Improving the urban energy system through local energy production:*
The elements on which the SEAP (Sustainable Energy Action Plan) focuses are the development of photovoltaics, the redevelopment of the district heating system and the deployment of small/medium distributed generation plants compatible with the protection of air quality.Supporting energy transition and circular economy processesThis implies the adoption of policies and practices that encourage the use of clean and sustainable energy, together with the encouragement of circular economy processes that minimize waste and maximize the reuse of resources.*Promoting renewable energy installations and the development of local distribution networks:* The Plan aims to replace the supply of gas and electricity from fossil sources with renewable energies for all needs. The development of local, integrated production and distribution systems powered by renewable energy sources (RESs) is proposed.*Encouraging the circular economy of construction and excavation materials*: The Plan aims to reduce land consumption and promote the circular economy for building materials. The city becomes a production site for recycled materials, reducing the use of nonrenewable resources and extending the life cycle of products.
*Increasing recycling and reduce waste production:*
The Plan proposes sustainable urban waste management with a network of recycling infrastructure. Buildings will be equipped with waste sorting spaces, and it is planned to expand existing waste sorting centers.Mobility planning focused on cycling and walkingThis goal implies the design and implementation of infrastructure and policies that encourage and promote cycling and walking as primary ways of transport in cities.
*Implementing cycling “Biciplan” routes:*
Definition of an integrated and extended project cycle network throughout the metropolitan territory as prefigured in the metropolitan “Biciplan”, classifying the network for daily mobility into strategic and integrative.*Promoting universal accessibility:* Creating a safe and inclusive pedestrian environment: always keeping the safety of the most fragile people at the center, the Urban Plan for Sustainable Mobility (UPSM) aims to plan and integrate the following strategies: pedestrian areas, pedestrian priority traffic zones, environmental restricted traffic zones, and the widespread adoption of the 30 km/h speed limit in many parts of the city (City 30).*Creating resilient public spaces:* Incorporating green infrastructure, such as permeable pavements or urban gardens, to manage stormwater and combat heat island effects.*Improving pedestrian accessibility to services* (“15 minute” cities)*Adopting the “Shared Space” approach:* Ensuring a general improvement in the perceptual conditions of safety and usability of spaces:The Plan proposes to reorganize public spaces in such a way as to promote safety and protect the means of active mobility, both in urban centers and on extra-urban provincial roads, in order to make them easier to cross and travel through.*Implementing “Smart City”*: Smart technologies to support safety and usability in cycling and walking.Upgrading and electrification of metropolitan public transportIt implies the implementation of policies and investments to improve and modernize the public transport system in urban areas, with a focus on its electrification.
*Defining a new metropolitan public transport network:*
The UPSM plans to offer a competitive alternative to private car use and to complement the metropolitan carrier network in a single integrated metropolitan fare system. The transport network envisages the following structure: carrier network, complementary network, and supplementary network.Network redistribution for vehicular mobilityIt refers to the change and reorganization of road and motorway networks to facilitate traffic circulation.*Promoting the “upgrading” of roads with a view to safety, quality of space, and landscaping* (30 zones, restricted traffic zones (RTZs), zero-emission zones (ZEZs) and other speed-avoidance strategies.It is suggested to optimize the use of existing resources and focus on improving the existing infrastructure instead of indiscriminate expansion of the road network. All projects should address all mobility components in an integrated planning perspective.Development of innovative mobility solutions.This implies the introduction of new technologies and ideas in the mobility sector, such as electric vehicles, autonomous vehicles, shared transport systems, and mobility-on-demand services.*Making Smart Mobility efficient and convenient alternatives to private vehicles available to the population*.Strengthen all possible sustainable modes and ensure their maximum integration. Thanks to new technologies in vehicles and mobility services, users will enjoy easy access to sustainable mobility, contributing to a reduction in the use of private motorized vehicles.Efficiency enhancement of public lighting installationsIt refers to the improvement of public lighting through the use of more efficient technologies, such as light emitting diode (LED) lamps, to reduce energy consumption and improve lighting quality.
*Replacing lighting fixtures with LED technology:*
The measure covers the city’s six districts uniformly and involves the replacement of all currently existing “yellow” light sources. By 2023, the experimental plan will lead to energy cost savings of approximately EUR 1 million.Control and regulation of public lighting systemIt refers to the implementation of systems and technologies to efficiently monitor and manage lighting systems used in public areas in order to optimize the energy efficiency of lighting installations.Equipment of intelligent control and regulation systems.Reductions in energy consumption, especially among particularly significant usersLarge reductions in energy consumption, especially among particularly significant users.Enhancement of green areas and the urban eco-networkIt aims to improve the salubrity of the urban environment and air quality by increasing green areas and implementing an ecological network in cities. This involves the creation of new parks, gardens, and green spaces, together with the promotion of ecological lanes that promote biodiversity and help reduce air pollution, referring to pollutants that affect both human health and the ecological balance of urban environments as particulate matter (PM2.5 and PM10), nitrogen oxides (NOx), volatile organic compounds (VOCs), and carbon dioxide (CO_2_).
*Implementing of green infrastructure within the city:*
The Plan recognizes the important role of permeable soils and green areas within the urban environment in regulating natural cycles, mitigating climate hazards, and supporting social and recreational services, especially in areas of high population density.
*Enhancing of periurban parks:*
The GAIA project is based on a public–private partnership model where private companies finance the purchase of plants and the maintenance of green spaces throughout the city, providing environmental benefits, in particular to mitigate the heat island effect. It provides for the planting of trees and urban gardens, as well as greening of buildings and public spaces.
*Safeguarding biodiversity and key hill and lowland ecosystem services:*
The Plan recognizes the crucial role of natural, renatured, and protected areas as vital reserves for biodiversity, regulation of natural cycles, and support for agricultural supply.*Tree planting within the urban area* (Action Plan for Sustainable Energy and Climate APSEC target: n 1300 trees in 10 years). Forestation in the rural area will be encouraged. The new parks (located in publicly owned areas) will contain equipped areas and areas for agricultural cultivation.Choose plant species with high environmental effectiveness:The aim is to combat climate change by planting trees, exploiting the biological functions of plants such as absorbing CO_2_ and purifying the air of pollutants.Prevention and mitigation of environmental risksIt refers to the implementation of measures and strategies to avoid or reduce negative impacts on the environment. The objective is to prevent environmental damage and risks to public safety from floods or water flows and to ensure sustainable management of water resources.*Ensuring hydraulic invariance:* For example, permeable pavements, green roofs, and rain gardens help absorb rainwater, reducing flood risks while preserving the area’s original water absorption capacity.
*Ensuring the consistent flow of water at the entrances of combined canals and ditches:*
The intervention takes place mainly on the hydraulic criticalities linked to the interferences largely referable to the covering of the hillside streams.*De-impermeabilizing the soil:* This refers to removing or reducing impermeable surfaces like concrete or asphalt to restore natural soil permeability.
*Identifying of areas at risk, rules and criteria for reducing vulnerability:*
The Plan deals with the prevention and reduction in risks, considering the dangerousness of events and the exposure of vulnerable elements to damage. In particular, it identifies areas characterized by different hazards (e.g., hydrogeological, hydraulic, seismic).
*Creating of a software platform to support the management of heavy rain events:*
The RainBo project aims to improve knowledge, methods, and tools for reacting to extreme rainfall events. It aims to achieve a monitoring framework based on a system of sensors and new technologies; an early-warning system; a system capable of simulating possible scenarios through hydrological models; a support tool for defining a response protocol to potential impacts.Developing “blue infrastructures”It refers to the creation and implementation of urban infrastructure that promotes the sustainable management of water resources, such as rivers and canals. The aim is to promote biodiversity and resilience to the effects of climate change.
*Building an urban blue infrastructure to facilitate the flow, purification, and absorption of water and to protect biodiversity:*
The Plan protects, enhances, and implements the blue infrastructure system. This not only promotes the flow, purification, and water retention, but also contributes to the protection of biodiversity, the reduction in air pollution (particulate matter (PM2.5 and PM10) and nitrogen oxides (NOx)), the reduction in energy demand, and the mitigation of the urban heat island effect.Learning, awareness, inclusionThe aim is to promote a culture of sustainability and inclusion to ensure significant community involvement in pursuing sustainable urban development goals.*Building regeneration models* (pilot case): In order to inspire broader urban regeneration efforts, these pilots raise awareness and demonstrate the viability of sustainable urban transformation.
*Building innovative participative tools for general interest decisions:*
The Bologna Climate Assembly is an instrument that directly involves citizens in making decisions of general interest. The Assembly’s task is to define recommendations to make Bologna a sustainable city by accelerating the energy transition.*Communicating environmental performance results:* Sharing data on sustainability efforts through reports, dashboards, assemblies.

### 3.4. Climate Action Matrix

All the data are collected and shown in Table 2.

## 4. Discussion

The research outlines a methodology for addressing climate change mitigation policies and targets in a key Italian municipality, which has been selected by the European Commission as one of 100 European cities, including 9 from Italy, to achieve climate neutrality by 2030. This methodology can be applied to other cities as well, offering a framework to evaluate, compare, and benchmark local policies against the broader European climate goals. In the context of the analyzed papers and implemented policies, it is crucial to invest in research, monitoring, and surveillance on climate change and public health to ensure a better understanding of the potential health co-benefits of climate mitigation at the local level. This paper represents a method to drive municipal policies towards dual-impact solutions aimed at both environmental and public health protection: it includes a support matrix targeted at policymakers and public administrations to support them in the implementation of policies into urban solutions.

It can be confirmed that the actions proposed by Bologna’s local policies anticipate the activities and obligations of the European Green Deal. All the measures reported contribute to varying degrees to mitigating climate change and protecting population health health.

Specifically, the assessment shows that some goals (1. Encourage the regeneration of manmade land and reduce land consumption, 2. Energy saving and efficiency in residential buildings, 3. Supporting energy transition and circular economy processes, 4. Mobility planning focused on cycling and walking, 5. Upgrading and electrification of metropolitan public transport, 6. Network redistribution for vehicular mobility, 7. Development of innovative mobility) are more directly involved than others. Moreover, some goals are quantitative, and others are qualitative. For instance, with respect to CO_2_ emissions reduction, the target indicators vary and range from 1700 tCO_2_ eq/year foreseen for goal 10 (Reductions in energy consumption, especially among particularly significant users n8) to 507.250 tCO_2_ eq/year expected for the first three goals belonging to the macro area of construction (Built Environment and Renewable Energy Sources). Some goals (12. Prevention and mitigation of environmental risks, 13. Developing “blue infrastructures”, and 14. Learning, awareness, inclusion) are not directly linked to the emissions reduction; for this reason, a quantitative indicator was not applicable.

On the other side, for example, the transport sector, that encourages active travel (e.g., walking and cycling), contributes effectively to the general objective as it produces significant reductions in cardiovascular disease, dementia, diabetes, obesity, and mental health improvements, affecting several risk urban factors, like air pollution, traffic injuries, and urban noise [2,30]. Therefore, considering what is reported by scientific literature and policies promoted by the Municipality of Bologna, as the implementation of cycling routes or the development of smart technologies to support safety, the city presents a progressive approach to foster public health, while integrating solutions for urban mobility problems.

Moreover, as outlined by the scientific literature [27] on public space, the enhancement of the urban ecosystem provides benefits for health, assisting in carbon sequestration: trees are particularly considered to be efficient in reducing concentrations of pollutants, especially particulate matter (PM2.5 and PM10) and carbon dioxide (CO_2_) Also, in this context, the Municipality of Bologna, due to the GAIA project [28], has increased green areas through the planting of at least 3000 trees, which is equivalent to 9000 tons of CO_2_ absorbed.

## 5. Conclusions

The research conducted here provides a foundational basis for the development of advanced methodologies aimed at influencing local policies that promote sustainable, health-promoting strategies. In terms of outcomes for urban populations, the environmental benefits are extensively documented and rigorously explored in the scientific literature, particularly through studies focusing on climate change, urban resilience, environmental challenges, and public health.

This study has some limitations that can be explored in future research. First, while the focus on the Bologna Municipality enabled an in-depth review of local policies, it represents a case study specific to this particular context, replicable for different contexts. Second, the research does not quantify the health impacts of individual interventions and policies to assess their public health benefits. Instead, it presents the effects primarily in terms of climate change mitigation and decarbonization, aligning with the European ambition of achieving urban carbon neutrality. At the same time, there are significant research gaps regarding the scientific evaluation of the health benefits of the mitigation actions. Therefore, future developments are needed; in particular, a statistical analysis on the direct health benefits is missing and could be introduced in follow-up studies. Interaction between public health experts, urban planners, designers, decision-makers and economists is thought to be effective. This study is a first step in supporting the transition toward multidisciplinary expertise building in implementing policies to mitigate climate change and promote public health in contemporary and future cities.

## Figures and Tables

**Figure 1 ijerph-21-01457-f001:**
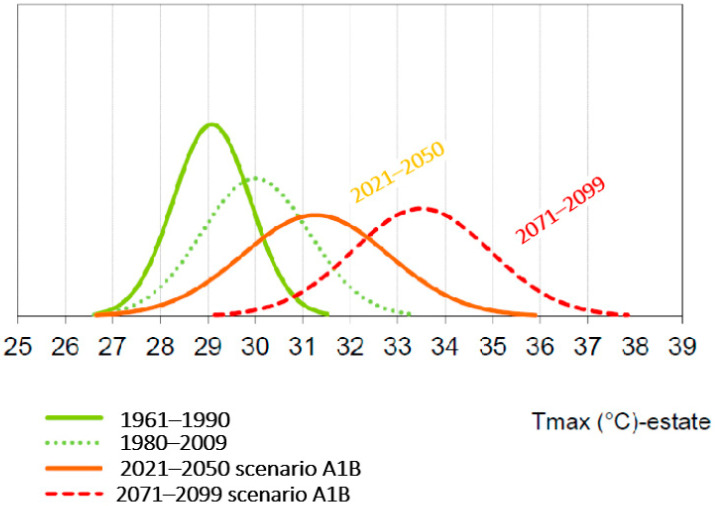
Analysis of the climate scenarios in the city of Bologna (elaborated), ARPAE.

**Figure 2 ijerph-21-01457-f002:**
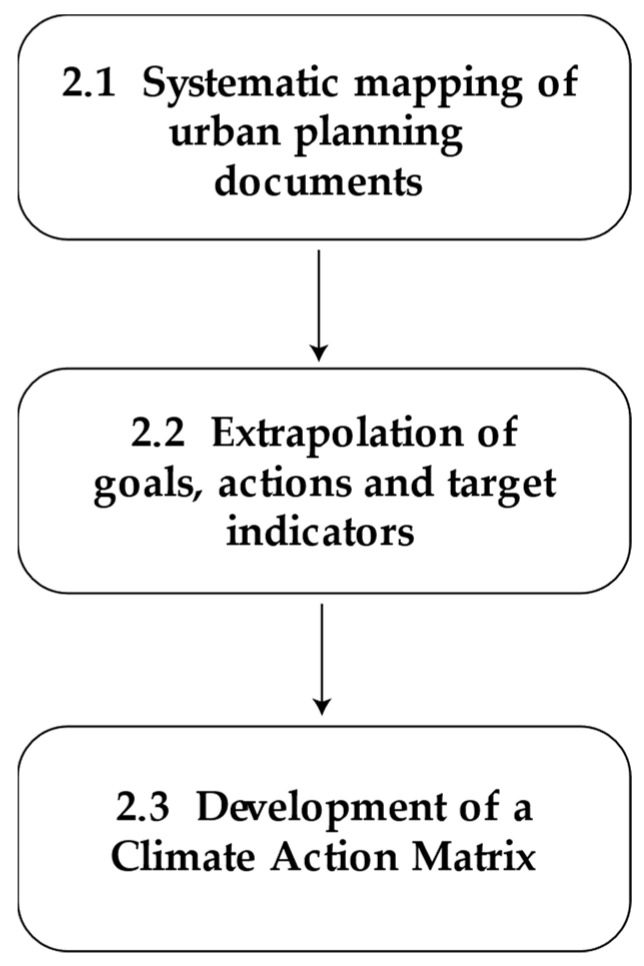
Methodological steps.

**Figure 3 ijerph-21-01457-f003:**
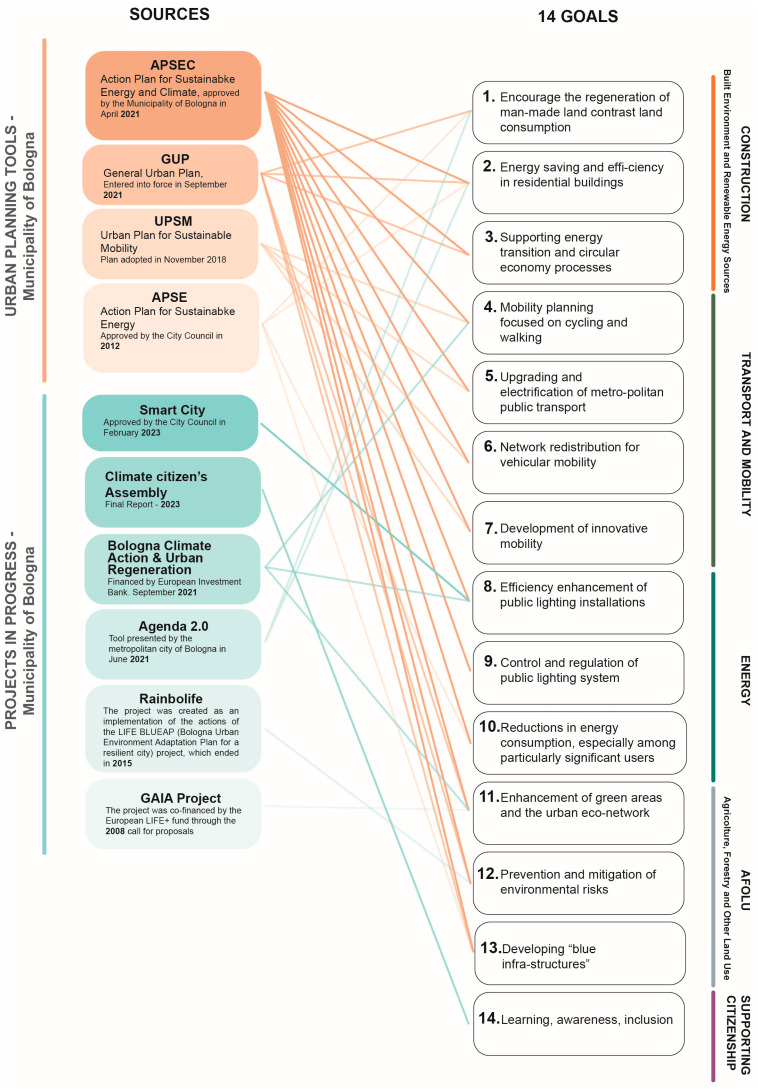
List of goals associated with the respective sources, divided by type.

**Table 1 ijerph-21-01457-t001:** Schematization of sources.

Name	Type	Source	Year	Citation
APSEC: Action Plan for Sustainable Energy and Climate	Urban planning tool	Municipality of Bologna	2021	[18]
GUP: General Urban Plan	Urban planning tool	Municipality of Bologna	2021	[19]
UPSM: Urban Plan for Sustainable Mobility	Urban planning tool	Metropolitan city of Bologna	2018	[20]
APSE: Action Plan for Sustainable Energy	Urban planning tool	Municipality of Bologna	2012	[21]
Smart City	Efficiency plan for public lighting installations	Municipality of Bologna	2023	[22]
Bologna Climate Assembly	Final report	Urban innovation foundation	2023	[23]
Bologna Climate Action & Urban Regeneration	Orientation framework of planning and programming instruments	Municipality of Bologna	2021	[24]
Agenda 2.0	Environmental project	Metropolitan city of Bologna	2021	[25]
Rainbolife	Environmental project	LIFE BLUEAP (Bologna Urban Environment Adaptation Plan for a resilient city)	2019	[26,27]
GAIA: Green Areas inner-city Agreement	Urban Forestation project	Municipality of Bologna	2008	[28]

**Table 2 ijerph-21-01457-t002:** Overall matrix for climate and sustainability strategies.

Goal	Measures/Actions	Target	Source
Encourage the regeneration of manmade land contrast land consumption.	Recovering and improving the efficiency of the existing building stock. Completing the parts of the city where transformation is not complete. Encourage reuse and urban regeneration of manmade areas.	CCC: Reduction of 507,250.13 tCO_2_ eq/year (Goals 1 + 2 + 3). APSEC: 222.342 tCO_2_ eq/year Equal to 9.7% compared to 2005 (Goals 1 + 2).	GUP: General Urban Plan, Municipality of Bologna, 2021 [19]. APSE. Action Plan for Sustainable Energy, Municipality of Bologna, 2012 [21]. Agenda 2.0, Metropolitan City of Bologna, 2021 [25].
2. Energy saving and efficiency in residential buildings.	Consolidating already existing functions. Define energy efficiency interventions for buildings. Promoting and incentivizing different forms of energy efficiency. Improvement of the urban energy system through local energy production.		APSEC: Action plan for Sustainable Energy and Climate, Municipality of Bologna, 2021 [18]. GUP: General Urban Plan, Municipality of Bologna, 2021 [19]. APSE. Action Plan for Sustainable Energy, Municipality of Bologna, 2012 [21]. Agenda 2.0, Metropolitan City of Bologna, 2021 [25].
3. Supporting energy transition and circular economy processes.	Promotion of renewable energy installations and the development of local distribution networks. Encouraging the circular economy of construction and excavation materials. Increase recycling and reduce waste production.	CCC: Reduction of 507,250.13 tCO_2_ eq/year (Goals 1 + 2 + 3). APSEC: RES + Hydrogen/biogas = Reduction of 159,637 tCO_2_ eq/year equal to 6.9% compared to 2005.	APSEC: Action plan for Sustainable Energy and Climate, Municipality of Bologna, 2021 [18]. GUP: General Urban Plan, Municipality of Bologna, 2021 [19].
4. Mobility planning focused on cycling and walking.	Cycling routes: implementation of metropolitan Biciplan routes. Promoting universal accessibility: creating a safe and inclusive pedestrian environment. Creation of resilient public spaces. Pedestrian accessibility to services (“15 minute” cities). “Shared Space” approach (reorganizing public spaces promoting safety). “Smart City”: smart technologies to support safety and usability in cycling and walking.	CCC: Reduction of 99,189.49 tCO_2_ eq/year (Goals 4 + 5 + 6 + 7). APSEC: Reduction of 110,051 tCO_2_ eq/year, equal to 4.8% compared to 2005 (Goals 4 + 5 + 6 + 7).	APSEC: Action plan for Sustainable Energy and Climate, Municipality of Bologna, 2021 [18]. UPSM: Urban Plan for Sustainable Mobility, 2018 [20]. Bologna Climate Action and Urban Regeneration, Municipality of Bologna, 2021 [24].
5. Upgrading and electrification of metropolitan public transport.	Definition of a new metropolitan public transport network.		APSEC: Action plan for Sustainable Energy and Climate, Municipality of Bologna, 2021 [18]. UPSM: Urban Plan for Sustainable [20].
6. Network redistribution for vehicular mobility.	Promoting the “upgrading” of roads with a view to safety, quality of space and landscaping (30 zones, ZTL, ZEZ and other speed-avoidance strategies).		
7. Development of innovative mobility.	Making Smart Mobility modes efficient and convenient alternatives to private vehicles.		
8. Efficiency enhancement of public lighting installations.	Replacement of lighting fixtures with LED technology.	APSEC: Reduction of 12,204 tCO_2_ eq/year, equal to 0.5% compared to 2005.	APSEC: Action plan for Sustainable Energy and Climate, Municipality of Bologna, 2021 [18]. APSE: Action plan for Sustainable Energy, Municipality of Bologna, 2012 [21].
9. Control and regulation of public lighting system.	Equipment of intelligent control and regulation systems.		APSEC: Action plan for Sustainable Energy and Climate, Municipality of Bologna, 2021 [18].
10. Reductions in energy consumption, especially among particularly significant users.	Large reductions in energy consumption, especially among particularly significant users.	CCC: Reduction of 2713 tCO_2_ eq/year. PAESC: Storage of 1700 tCO_2_ eq/year.	APSEC: Action plan for Sustainable Energy and Climate, Municipality of Bologna, 2021 [18]. GUP: General Urban Plan, Municipality of Bologna, 2021 [19]. Bologna Climate Action and Urban Regeneration, Municipality of Bologna, 2021 [24]. GAIA Project, European call, 2008 [28].
11. Enhancement of green areas and the urban eco-network.	Implementation of green infrastructure within the city. Enhancement of periurban parks. Preserving biodiversity and the main ecosystem services. Tree planting within the urban area (PAESC target: n 1300 trees in 10 years). Choose plant species with high environmental effectiveness.	APSEC: Equip at least 1% of built/paved land with sustainable drainage systems. Facilitate the removal of 50% of the pollutant load delivered by surface waters.	APSEC: Action plan for Sustainable Energy and Climate, Municipality of Bologna, 2021 [18]. GUP: General Urban Plan, Municipality of Bologna, 2021 [19]. RAINBOLIFE, 2015 [27,28].
12. Prevention and mitigation of environmental risks.	Ensuring hydraulic invariance Ensure regular water flow in the mouths of tombed streams and ditches. Soil de-impermeabilization. Identification of risk areas, rules and criteria for reducing vulnerability. Creation of a software platform to support the management of heavy rain events.	n/a (not applicable).	APSEC: Action plan for Sustainable Energy and Climate, Municipality of Bologna, 2021 [18]. GUP: General Urban Plan, Municipality of Bologna, 2021 [19].
13. Developing “blue infrastructures”.	Building an urban blue infrastructure to facilitate the flow, purification, and absorption of water and to protect biodiversity.	n/a (not applicable).	Climate citizen’s Assembly, 2023 [23].
14. Learning, awareness, inclusion.	Building regeneration models (pilot case). Building innovative participative tools for general interest decisions. Communication of environmental performance results.	n/a (not applicable).	Climate citizen’s Assembly, 2023 [23].

## Data Availability

The original data presented in the study are openly available through the Bologna Municipality website: www.comune.bologna.it.

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
