# Peer review of "Integrated Climate Change Mitigation and Public Health Protection Strategies: The Case of the City of Bologna, Italy"

_ijerph, 2024, doi:10.3390/ijerph21111457_

Round 1
Reviewer 1 Report
Comments and Suggestions for Authors
This manuscript systematically analyzes the policies and measures of Bologna in addressing climate change and protecting public health, providing a detailed case study. The research question is meaningful, proposing specific practical strategies and matrices. However, it is crucial to underscore the potential value and insights that the case of Bologna City can offer to a global audience of international readers. There is still a way to go before it becomes a academic paper. It is more like a research report at the moment.
Introduction:
In the introduction section, the manuscript provides a detailed explanation of the current situation and policies regarding carbon neutrality in Europe, but there is limited background information specifically related to the city of Bologna, Italy.
Page 3 Line 110 - Enhancing the analysis of specific socio-economic conditions, environmental status, and historical climate data in the city of Bologna is valuable to highlight the practical significance of the research.
Data:
Page 4 Line 135 - The data here has only been classified; a detailed description of the data processing process can be provided.
Discussion:
The Discussion section lacks depth and requires a more thorough examination. You need to compare the manuscript with existing literature, discussing what contribution has been done. Additionally, some of the content in the Conclusions should be integrated into the Discussion while keeping the Conclusion brief.
Page 13, line 427, it explains the importance of policies such as encouraging travel but lacks discussing other relevant policies. Suggestions can be provided along with an in-depth analysis that incorporates international policies and practical experiences.
Author Response
Comment 1: [ In the introduction section, the manuscript provides a detailed explanation of the current situation and policies regarding carbon neutrality in Europe, but there is limited background information specifically related to the city of Bologna, Italy. ]
Response 1: [In 2015 the inventory of emissions was updated to assess the trends of the city and monitor the effectiveness of what was implemented, recording a reduction of almost 300.000 tons of CO2, or a drop of 12,4%. This is a very positive result that outlines a new trend that will probably allow us to achieve the -20% target to 2020. At the same time, Bologna was one of the first cities in Italy to adopt a local plan for adaptation to climate change (Fini et al., 2016). This Plan is the result of the BLUE AP project (Bologna Local Urban Environment Adaptation Plan for a Resilient City), funded by the LIFE+ program (LIFE11 ENV/IT/119), which the Municipality of Bologna coordinated between 2012 and 2015 involving Kyoto Club, Ambiente Italia, ARPAE Emilia Romagna and CMCC (Euro-Mediterranean Center for Climate Change), (Boeri et al., 2018)]. + Fig.1. Analysis of the climate scenarios in the city of Bologna (elaborated).
Comment 2 [Page 3, Line 110 - Enhancing the analysis of specific socio-economic conditions, environmental status, and historical climate data in the city of Bologna is valuable to highlight the practical significance of the research.]
Comment 2 [The analysis has been enhanced: Bologna’s climate data, as many other European cities, underscored the need for climate adaptation measures. Bologna is experiencing improvements in air quality partially due to stricter emissions standards and expansion of pedestrian zone. Interventions in affordable housing and public services are needed to ensure equitable access for all socioeconomic groups].
Comment 3: [Page 4, Line 135 - The data here has only been classified; a detailed description of the data processing process can be provided]
Response 3: [In defining objectives and indicators, first it has been done a general consultation of tools related to the environmental aspect of the Municipality of Bologna; successively, the relevant documents have been gathered and categorized into two main groups: urban planning tools and ongoing projects. While the first ones include formal documents ad master plans and transportation plans, the second ones are more specific projects, detailing current infrastructure development or social initiatives being undertaken by the Municipality of Bologna].
Comment 4: [The Discussion section lacks depth and requires a more thorough examination. You need to compare the manuscript with existing literature, discussing what contribution has been done. Additionally, some of the content in the Conclusions should be integrated into the Discussion while keeping the Conclusion brief]
Response 4: [The first paragraph of the old Conclusion has been replaced in the Discussion]
Comment 5: [Page 13, line 427, it explains the importance of policies such as encouraging travel but lacks discussing other relevant policies. Suggestions can be provided along with an in-depth analysis that incorporates international policies and practical experiences.]
Response 5: [The paragraph has been updated as follows, also integrating results from a literature review addressing active travel as health promoting solution:
De Nazelle, Audrey, Mark J. Nieuwenhuijsen, Josep M. Antó, Michael Brauer, David Briggs, Charlotte Braun-Fahrlander, Nick Cavill, et al. 2011. “Improving Health through Policies That Promote Active Travel: A Review of Evidence to Support Integrated Health Impact Assessment.” Environment International 37 (4): 766–77. https://doi.org/10.1016/j.envint.2011.02.003.
On the other side, for example, the transport sector, that encourage active travel (e.g., walking and cycling), contribute effectively to the general objective as it produces significant reductions in cardiovascular disease, dementia, diabetes, obesity and mental health improvements, affecting several risk urban factors, like air pollution, traffic injuries, and urban noise [2, 30]. Therefore, considering what is reported by scientific literature and the policies promoted by the Municipality of Bologna, as the implementation of cycling routes or the development of smart technologies to support safety, it can be said that the city presents a progressive approach to foster Public Health, while integrating solutions for urban mobility problems.]

Reviewer 2 Report
Comments and Suggestions for Authors
While the manuscript presents a significant topic, there are several areas that could benefit from improvement. Below, I have outlined specific suggestions:
Title
- Please include the country of the study area in the title for clarity.
Abstract
- Ensure consistent use of capital letters throughout the abstract.
- Include key details such as the study type, study period, country of the study area, and a brief description of the data analysis method.
Introduction
- Check the running order of references throughout the manuscript to ensure accuracy.
- Line 54, when mentioning the “spread of diseases”, please specify and elaborate on the diseases being referred to.
- Line 94, provide the country of the study area for clarity and context.
Materials and Methods
- Provide the study type, study period, and details on data collection, including inclusion criteria, exclusion criteria, keywords, and the databases used. Also, outline the process for data extraction and data synthesis.
- Ensure that the figure is properly cited within the text.
Results
- Verify that the reference numbers in the text correspond correctly with Table 1.
- Ensure that the figure is accurately cited within the text.
- In line 262, clarify the abbreviation “RES” by providing its full name. Similarly, in line 282, identify the full name for “PUMS”.
- For line 286-287, provide additional explanation or context to enhance understanding.
- In line 309, provide the full names for the abbreviations “ZTL” and “ZEZ”. In line 326, also clarify what “LED” stands for.
- In line 345, specify the types of “air pollution” being referred to.
- In line 362, provide the full name for the abbreviation “PAESC”.
- In line 367, verify the chemical formula for “CO2” to ensure accuracy.
- In line 374, 379, 406, and 412, provide additional details or explanations for clarity.
- In line 401, specify the type of “air pollution” being referred to.
- In Table 2, ensure the table is cited appropriately in the text.
- The manuscript title pertains to public health; however, the findings do not seem to reflect this focus. Please clarify how the findings relate to public health.
Discussion
- In line 419, 420, and 425, specify which “some goals” are being referred to.
- In line 435, identify the specific “pollutants” mentioned.
- Expand the discussion to include more detail on the limitations of the study, provide recommendations for further research, and offer recommendations for relevant agencies.
Conclusions
- Ensure this section does not repeat content from the Results. Keep it precise and concise, and avoid citing any references.
References
- Verify that the number of references matches those cited in the text and ensure their accuracy and correctness.
Author Response
Comment 1: Please include the country of the study area in the title for clarity.
Response 1: Done
Comment 2: Ensure consistent use of capital letters throughout the abstract. Include key details such as the study type, study period, country of the study area, and a brief description of the data analysis method.
Response 2: Done and included. Introduction: The ongoing global warming, driven by the escalating concentration of greenhouse gases from human activities, especially in urban areas, significantly impacts Public Health. Contemporary cities play an important role in health promotion and disease prevention, and some have the target to achieve net-zero Greenhouse Gas emissions shortly. There is a consistent guide for reaching this goal and there is the need to map and synthetize effective strategies and regulations. Materials and Methods: The study consists in the analysis of policy guidelines of the city of Bologna, consulted in March and April 2024. The following methodology was adopted for the analysis of climate change mitigation strategies in the city of Bologna case study, one of the 100 cities that will work towards achieving climate neutrality by 2030, 20 years ahead of the EU target: i) systematic mapping of sources and spatial planning documents; ii) extrapolation of goals, measures and target indicators; iii) development of an overall matrix.. Results: The main findings of the study and their connection to Public Health are: the identification of key macro-areas contributing to the reduction of greenhouse gas emissions, while reducing the impact of climate change on Health: 1. Building, 2. Built Environment and Renewable Energy Sources; 3. Transport and Mobility; Energy; Green Areas and Land Use; Citizen Support. Within these 5 macroareas, 14 goals have been identified, to which a total of distinct 36 measures correspond and, finally, a target indicator is associated, mainly with respect to the reduction of tons of CO2 equivalent per year. Conclusions: With a view of protecting Public Health, it emerges that buildings and urban activities should not produce carbon emissions throughout their lifecycle. The paper presents a method to evaluate municipal policies towards dual-impact solutions that addresses both environmental protection through sustainability strategies and Public Health, aligned with the Health in All Policies (HiAP) approach
Comment 3: Introduction: a. Check the running order of references throughout the manuscript to ensure accuracy. Line 54, when mentioning the “spread of diseases”, please specify and elaborate on the diseases being referred to. Line 94, provide the country of the study area for clarity and context.
Response 3: a. OK, B. It’s scientifically proven that Non-Communicable Disease (NCDs) – like cardio-respiratory and skin diseases, cancer, allergopathies, obesity, diabetes, stress, anxiety, sleeping disorders, cognitive development and social exclusion – are mainly connected to the Environmental Risk Factor [25]. C. Ok
Comment 4: Materials and Methods. Provide the study type, study period, and details on data collection, including inclusion criteria, exclusion criteria, keywords, and the databases used. Also, outline the process for data extraction and data synthesis. Ensure that the figure is properly cited within the text.
Response 4: As integrated in the text, the study consists in the analysis of policy guidelines of the city of Bologna, consulted in March and April 2024 from the site of the Municipality and of the Metropolitan City of Bologna. Figures have been corrected.
Comment 5: Results: Verify that the reference numbers in the text correspond correctly with Table 1; Ensure that the figure is accurately cited within the text.; In line 262, clarify the abbreviation “RES” by providing its full name. Similarly, in line 282, identify the full name for “PUMS”; For line 286-287, provide additional explanation or context to enhance understanding.; In line 309, provide the full names for the abbreviations “ZTL” and “ZEZ”. In line 326, also clarify what “LED” stands for.; In line 345, specify the types of “air pollution” being referred to it; In line 362, provide the full name for the abbreviation “PAESC”.; In line 367, verify the chemical formula for “CO2” to ensure accuracy.; In line 374, 379, 406, and 412, provide additional details or explanations for clarity.; In line 401, specify the type of “air pollution” being referred to.; In Table 2, ensure the table is cited appropriately in the text.; The manuscript title pertains to public health; however, the findings do not seem to reflect this focus. Please clarify how the findings relate to public health.
Response 5:
- In line 262, clarify the abbreviation “RES” by
providing its full name. Similarly, in line 282, identify the
full name for “PUMS”. Renewable Energy Sources. Urban Plan for Sustainable Mobility (UPSM)
- For line 286-287, provide additional explanation
or context to enhance understanding.
Incorporating green infrastructure, such as permeable pavements or urban gardens, to manage storm water and combat heat island effects.
- In line 309, provide the full names for the
abbreviations “ZTL” and “ZEZ”. In line 326, also clarify
what “LED” stands for.
Restricted Traffic Zones RTZ, Zero Emission Zone ZEZ.
- In line 345, specify the types of “air pollution”
being referred to.
Referring to pollutants that affect both human health and the ecological balance of urban environments as Particulate Matter (PM2.5 and PM10), Nitrogen Oxides (NOx), Volatile Organic Compounds (VOCs) and Carbon Dioxide (CO2).
- In line 362, provide the full name for the
abbreviation “PAESC”.
Action Plan for Sustainable Energy and Climate APSEC
- In line 367, verify the chemical formula for “CO2”
to ensure accuracy. CO2
- In line 374, 379, 406, and 412, provide additional
details or explanations for clarity.
- Ensuring hydraulic invariance. for example, permeable pavements, green roofs, and rain gardens help absorb rainwater, reducing flood risks while preserving the area's original water absorption capacity.
- De-impermeabilizing the soil: refers to removing or reducing impermeable surfaces like concrete or asphalt to restore natural soil permeability.
- Building regeneration models (pilot case): to inspire broader urban regeneration efforts., these pilots raise awareness and demonstrate the viability of sustainable urban transformation.
- Communicating environmental performance results: sharing data on sustainability efforts through reports, dashboards, assemblies.
Comment 6: Discussion: A. In line 419, 420, and 425, specify which “some goals” are being referred to. B. In line 435, identify the specific “pollutants” mentioned. C. Expand the discussion to include more detail on the limitations of the study, provide recommendations for further research, and offer recommendations for relevant agencies.
Response 6: A. Specifically, the assessment shows that some goals (1. Encourage the regeneration of man-made land contrast land consumption, 2. Energy saving and efficiency in residential buildings, 3. Supporting energy transition and circular economy processes, 4. Mobility planning focused on cycling and walking, 5. Upgrading and electrification of metropolitan public transport, 6. Network redistribution for vehicular mobility, 7. Development of innovative mobility) are more directly involved than others that are less influential. […]
C. In the context of the analyzed papers and implemented policies, it can be stated that it becomes crucial to invest in research, monitoring and surveillance on climate change and Public Health to ensure a better understanding of the potential health co-benefits of climate mitigation at local level. With a view of protecting Public Health, cities long-term vision is that buildings should not produce carbon emissions throughout their lifecycle. This paper represents a method to drive municipal policies towards dual-impact solutions aimed at both environmental and Public Health protection: it consists of a support matrix targeted at Policy Makers and Public Administrations in order to support them in the implementation of policies into urban solutions. The study has some limitations that can be explored in future research. First, while the focus on the Bologna municipality enabled an in-depth review of local policies, it represents a case study specific to this particular context, which can be then replicated to different context. Second, the research does not quantify the health impacts of individual interventions and policies to assess their public health benefits. Instead, it presents the effects primarily in terms of climate change mitigation and decarbonization, aligning with the European ambition of achieving urban carbon neutrality.
Comment 7: Ensure this section does not repeat content from the Results. Keep it precise and concise, and avoid citing any references.
Response 7: OK.
Comment 8: References: Verify that the number of references matches those cited in the text and ensure their accuracy and correctness.
Response 8: OK.

Reviewer 3 Report
Comments and Suggestions for Authors
This article investigates how urban planning and policy initiatives in Bologna align with climate neutrality goals while promoting public health. Specifically, the study focuses on the city's strategies for reducing greenhouse gas emissions and adapting to climate change through a "Health in All Policies" approach. It provides a solid framework for analyzing the integration of climate mitigation and public health strategies at the municipal level. However, it requires substantial revision to warrant publication as a high-quality research article. Detailed comments follow.
Abstract: I would expect the authors to highlight the main contributions of this study. Currently it is too long. Please streamline the current version. Also, the findings are scattered. What are the main findings of this study that can significantly contribute to the exitsing studies in this field?
The introduction section covers some of the literature in this field but still lacks enough literature to justify the current gaps in the literature. It is recommended that the authors read and cite the following important literature: 1) Motivating mitigation: when health matters more than climate change." Climatic Change. 2) Government response to climate change in China: A study of provincial and municipal plans. Journal of Environmental Planning and Management. 3) How resilient are localities planning for climate change? An evaluation of 50 plans in the United States. Journal of Environmental Management.
Research method: this session mentions that the evaluation matrix does not include direct indicators, which limits the quality of the analysis. To strengthen the paper, some statistical analysis may be incorporated.
Discussion: Currently it is a bit scattered. I would expect the authors to include more in-depth discussion in this session. What is the most innovative aspect of this research?
Last but not least, it is recommended that the authors perform a thorough edit to ensure proper grammar and clear presentation. Currently, there are persistent grammatical errors and non-standard citations throughout the manuscript.
Comments on the Quality of English LanguageThe quality of English language is generally fine, but needs to be extensively revised.
Author Response
Comment 1: Abstract: I would expect the authors to highlight the main contributions of this study. Currently it is too long. Please streamline the current version. Also, the findings are scattered. What are the main findings of this study that can significantly contribute to the existing studies in this field?
Response 1: Abstract has been adjusted according to the indications provided
Comment 2: The introduction section covers some of the literature in this field but still lacks enough literature to justify the current gaps in the literature. It is recommended that the authors read and cite the following important literature: 1) Motivating mitigation: when health matters more than climate change." Climatic Change. 2) Government response to climate change in China: A study of provincial and municipal plans. Journal of Environmental Planning and Management. 3) How resilient are localities planning for climate change? An evaluation of 50 plans in the United States. Journal of Environmental Management.
Response 2: Publications suited the scope of the paper, and therefore have been integrated as references
Comment 3: Research method: this session mentions that the evaluation matrix does not include direct indicators, which limits the quality of the analysis. To strengthen the paper, some statistical analysis may be incorporated.
Response 3: See paragraph 2.2. Extrapolation of Goals, Actions and Target Indicators
Comment 4: Discussion: Currently it is a bit scattered. I would expect the authors to include more in-depth discussion in this session. What is the most innovative aspect of this research?
Response 4: The research outlines a methodology for addressing climate change mitigation policies and targets in a key Italian municipality, which has been selected by the European Commission as one of 100 European cities, including 9 from Italy, to achieve climate neutrality by 2030. This methodology can be applied to other cities as well, offering a framework to evaluate, compare, and benchmark local policies against the broader European climate goals. In the context of the analyzed papers and implemented policies, it can be stated that it becomes crucial to invest in research, monitoring and surveillance on climate change and Public Health to ensure a better understanding of the potential health co-benefits of climate mitigation at local level. This paper represents a method to drive municipal policies towards dual-impact solutions aimed at both environmental and Public Health protection: it consists of a support matrix targeted at Policy Makers and Public Administrations in order to support them in the implementation of policies into urban solutions
Comment 5: Last but not least, it is recommended that the authors perform a thorough edit to ensure proper grammar and clear presentation. Currently, there are persistent grammatical errors and non-standard citations throughout the manuscript.
Response 5: English as been revised by a mother-tongue colleague, but if needed, we are willing to ask for the revision service of the editor.

Round 2
Reviewer 1 Report
Comments and Suggestions for Authors
I have no further comments.
Reviewer 3 Report
Comments and Suggestions for Authors
The authors have adequately addressed my previous concerns.